# SGLT2 Inhibitors: A Review of Their Antidiabetic and Cardioprotective Effects

**DOI:** 10.3390/ijerph16162965

**Published:** 2019-08-17

**Authors:** Anastasios Tentolouris, Panayotis Vlachakis, Evangelia Tzeravini, Ioanna Eleftheriadou, Nikolaos Tentolouris

**Affiliations:** Diabetes Center, First Department of Propaedeutic Internal Medicine, Medical School, National and Kapodistrian University of Athens, Laiko General Hospital, 11527 Athens, Greece

**Keywords:** type 2 diabetes, sodium-glucose co-transporter 2 inhibitors, glycemic control, cardiovascular disease, cardioprotective mechanisms, adverse effects, empagliflozin, dapagliflozin, canagliflozin, ertugliflozin

## Abstract

Type 2 diabetes mellitus is a chronic metabolic disease associated with high cardiovascular (CV) risk. Sodium-glucose co-transporter 2 inhibitors (SGLT2i) are the latest class of antidiabetic medication that inhibit the absorption of glucose from the proximal tubule of the kidney and hence cause glycosuria. Four SGLT2i are currently commercially available in many countries: canagliflozin, dapagliflozin, empagliflozin, and ertugliflozin. SGLT2i reduce glycated hemoglobin by 0.5%–1.0% and have shown favorable effects on body weight, blood pressure, lipid profile, arterial stiffness and endothelial function. More importantly, SGLT2i have demonstrated impressive cardioprotective and renoprotective effects. The main mechanisms underlying their cardioprotective effects have been attributed to improvement in cardiac cell metabolism, improvement in ventricular loading conditions, inhibition of the Na^+^/H^+^ exchange in the myocardial cells, alteration in adipokines and cytokines production, as well as reduction of cardiac cells necrosis and cardiac fibrosis. The main adverse events of SGLT2i include urinary tract and genital infections, as well as euglycemic diabetic ketoacidosis. Concerns have also been raised about the association of SGLT2i with lower limb amputations, Fournier gangrene, risk of bone fractures, female breast cancer, male bladder cancer, orthostatic hypotension, and acute kidney injury.

## 1. Introduction

Diabetes mellitus (DM) is a common metabolic disease characterized by chronic hyperglycemia that globally affects approximately 10% of the population, and its prevalence is expected to increase in the following years [1]. The leading cause of morbidity and mortality in the diabetic population remains cardiovascular (CV) disease, which is estimated to be 2-4 times more common in patients with DM when compared with people without DM [2,3].

Tight glycemic control with the use of older antidiabetic agents such as metformin, thiazolidinediones (TZD), sulfonylureas (SU) and meglitinides, as well as the dipeptidyl peptidase inhibitors (DPP4i), has been associated with reductions in microvascular complications, however, no significant effect was observed on macrovascular disease [4,5,6]. On the other hand, the two latest classes of antidiabetic medication, glucagon peptide-1 receptor agonists (GLP1-RA) and sodium-glucose co-transporter (SGLT) 2 inhibitors (SGLT2i) have demonstrated cardioprotective effects and reduction in CV outcomes [6]. The latest guidelines of the American Diabetes Association (ADA) for the management of hyperglycemia recommend a personalized approach [7]. CV disease and the presence of chronic kidney disease (CKD) are the main comorbidities that affect the selection of the second agent added on metformin, which remains the first treatment option for every patient with type 2 DM (T2DM) [7].

SGLT2i inhibit the absorption of glucose from the proximal tubule of the kidney and hence cause glycosuria. The inhibition of SGLT2 is not a novel approach since it was introduced 130 years ago by Belgian and French scientists [8]. At present, four oral agents are approved for the treatment of T2DM by the U.S. Food and Drug Administration (FDA) and the European Medicines Agency (EMA): canagliflozin (Cana), dapagliflozin (Dapa), empagliflozin (Empa) and ertugliflozin (Ertu).

The aim of this review is to present the antidiabetic and cardioprotective profile of SGLT2i based on safety trials, randomized controlled trials (RCT), meta-analyses, large cohort studies, and real-world data. We used the keywords “canagliflozin, dapagliflozin, empagliflozin, ertugliflozin, type 2 diabetes, sodium-glucose co-transporter 2 inhibitors, glycemia, cardiovascular disease, cardioprotective mechanisms, adverse effects” alone and in combination to retrieve available literature data from PubMed between the years 2010 and 2019.

## 2. Sodium Glucose Co-Transporter Inhibition

### 2.1. Sodium-Glucose Co-Transporter 1 and 2

Under normal conditions, all filtered glucose undergoes reabsorption in the kidney tubules and therefore, no glucose is present in the urine [9]. SGLT2s located in the early S1 segment of the proximal tubule are responsible for the reabsorption of 80%–90% of filtered glucose, while SGLT1s located in the S2/S3 segment of proximal tubule reabsorb the remaining 10%–20% [10,11]. Thus, glucose that escapes SGLT2 is reabsorbed by SGLT1 in more distal tubular segments. Apart from the proximal tubule epithelium, SGLT2s are located in pancreatic a-cells and in the cerebellum, while SGLT1s are more widely distributed to kidneys, intestine, heart, lungs and skeletal muscles [12,13].

The maximum renal glucose reabsorption capacity is 375 mg per minute that corresponds to glucose levels of 300 mg/dL, while the rate at which glucose is filtered through the glomerulus is approximately 125 mg per minute [10,14]. Therefore, under normal circumstances, all filtered glucose is reabsorbed. However, when plasma glucose concentration exceeds 180 mg/dL, filtered glucose is excreted in the urine. This threshold is referred to as the “threshold for glycosuria” [10]. The difference between the actual threshold of 180 mg/dL and the theoretical threshold of 300 mg/dL is referred to as “splay” and is explained by the presence of functional and/or morphological glomerulotubular imbalance [10]. A decrease in the threshold of glycosuria, an increase in splay or a reduction in renal glucose reabsorption capacity lead to glycosuria [10].

Studies have shown that people with DM have elevated levels of renal threshold for glycosuria and increased renal glucose reabsorption capacity, and this applies even for patients with good glycemic control [10,15]. This increased activity is mainly due to the increased expression of SGLT2s in tubular epithelial cells in patients with DM when compared with people without DM [16]. This mechanism represents an adaptation of the human body to prevent energy loss and it seems to contribute to the multifunctional pathogenesis of hyperglycemia in subjects with DM [10].

### 2.2. Sodium-Glucose Co-Transporter 1 and 2 Inhibition

Phlorizin was isolated from apple trees in 1835 and is the first natural SGLT inhibitor that has high-affinity, specific, and competitive inhibitory activity for both SGLT1 and SGLT2 [17,18]. At first, phlorizin was used for the treatment of fever, malaria and other infectious diseases, however, within years it was discovered that phlorizin causes glycosuria. Several phlorizin analogues have been developed with different potency and selectivity against SGLT. In 2008, Dapa was developed that has more than 1200-fold higher potency for SGLT2 than SGLT1 [19]. Cana is another phlozin derivative with 400-fold higher inhibitory activity for SGLT2 than for SGLT1 [20]. The third agent of this class was Empa, which has the highest selectivity for SGLT2 over SGLT1 (approximately 2700-fold) among the commercially available SGLT2i [21]. The fourth phlorizin analogue developed was Ertu, which has 2200-fold higher selectivity for SGLT2 than for SGLT1 [22]. Dapa was approved by the EMA in 2012 and by the FDA in 2014, Cana was approved by both the EMA and FDA in 2013, and Empa in 2014 [17]. Ertu was approved by the FDA in 2017 and by the EMA in 2018 [17].

In the market, tablets of Dapa 5 mg and 10 mg, Empa 10 mg and 25 mg, Cana 100 mg and 300 mg, and Ertu 5 mg and 15 mg are available in many countries [10,11,17]. It should be noted that full doses of Empa (25 mg) and Cana (300 mg) can be administered when the estimated glomerular filtration rate (eGFR) is over 60 mL/min/1.73 m^2^ [10,11,17]. When the eGFR is between 45–60 mL/min/1.73 m^2^, a lower dose (10 mg and 100 mg, respectively) is recommended to those who already receive them and do not experience adverse effects. When the eGFR is below 45 mL/min/1.73 m^2^, the treatment should be stopped. Dapa can be administered only when the eGFR is over 60 mL/min/1.73 m^2^. Ertu should not be initiated in patients with an eGFR below 60 mL/min/1.73 m^2^ and should be discontinued when eGFR is persistently below 45 mL/min/1.73 m^2^ [10,11,17].

## 3. Glycemic Efficacy of Sodium-Glucose Co-Transporter 2 Inhibitors (SGLT2i)

All new antidiabetic agents have to present effective results on glycemic control to apply for marketing authorization. Glycemic control is usually examined through the glycated hemoglobin (HbA1c) levels, as well as through fasting plasma glucose (FPG) and postprandial glucose (PPG) levels. The SGLT2i have completed a series of phase III, double blind, placebo-controlled trials that examine the glycemic effect when added on either treatment-naive patients, or patients treated with metformin, SU, TZD, DPP4i, GLP1-RA or basal insulin as monotherapy.

### 3.1. Dapagliflozin (Dapa)

Firstly, the effect of Dapa on glycemic control was examined in treatment-naïve patients with T2DM [23]. After 24 weeks of intervention, the administration of Dapa 1, 2.5 and 5 mg once daily resulted in a reduction of HbA1c by 0.68%, 0.72%, and 0.82%, respectively [23]. At the same time, Dapa led to reduced FPG levels in comparison with placebo. These findings were observed in another study with similar protocol: HbA1c was reduced by 0.58%, 0.77%, and 0.89% with the administration of 2.5, 5 and, 10 mg Dapa, respectively [24]. Interestingly, when patients with higher baseline HbA1c (10.1%–12.0%) were treated with Dapa, greater reductions in HbA1c levels were induced (−2.88% with 5 mg and −2.66% with 10 mg) [24]. Therefore, the level of change in HbA1c is strongly associated with baseline HbA1c values.

In addition, several short-term studies and their long-term extensions have examined the glycemic effect of Dapa in patients who were already treated with metformin [25,26,27,28,29]. Indicatively, Bailey et al. randomized 546 subjects to Dapa 2.5, 5, 10 mg or placebo once daily for a period of 24 weeks [25]. Treatment with Dapa resulted in greater reductions in HbA1c (−0.67% with 2.5 mg, −0.70% with 5 mg and −0.84% with 10 mg) when compared with placebo [25]. In this study, a subgroup of participants received Dapa in the evening and the results were similar to those obtained by morning treatment, suggesting that the time of administration does not affect Dapa’s action. The glucose-lowering effect of Dapa was sustained in the 78-week extension of the trial [26]. A meta-analysis assessed the change in HbA1c when TZDs, SUs, DDP4i or Dapa were administered in people with T2DM who were inadequately controlled with metformin monotherapy [30]. The mean change in HbA1c from baseline was similar across the different agents. The impact of Dapa on HbA1c reduction was −0.08% relative to the one induced by DPP4i and −0.02% relative to the one induced by TZDs, while similar HbA1c reduction was observed when compared with treatment with SUs [30]. In terms of hypoglycemic events, TZDs, DDP4i, and Dapa were associated with a reduced risk of hypoglycemia when compared with SUs [30].

Dapa has been also examined as add-on therapy on glimepiride [31]. After 24 weeks of intervention, HbA1c was reduced by 0.58%, 0.63%, 0.82% and 0.13% with Dapa 2.5, 5, 10 mg and placebo, respectively [31]. In addition, 2 h PPG in response to an oral glucose tolerance test was reduced by 32 and 35 mg/dL with Dapa 5 and 10 mg respectively, whereas FPG was reduced by 21 and 28 mg/dL, respectively. Hence, this study confirmed that Dapa has both fasting and postprandial efficacy which is essential for optimal glycemic control. Although Dapa does not cause hypoglycemia, hypoglycemic events were reported more frequently in the Dapa groups (6.9%–7.9%) than in the placebo group (4.8%), most likely due to the combination of improved glycemic control with Dapa and the hypoglycemic effects of glimepiride [31]. Nevertheless, no patient discontinued the study as a result of hypoglycemia. These findings were sustained in a 24-week extension of this trial [32].

Another study examined the glycemic effect of Dapa when added in patients with T2DM inadequately controlled with pioglitazone [33]. HbA1c was reduced in the Dapa treatment arm by 0.82% and 0.97% with 5 and 10 mg of Dapa respectively, after 24 weeks of treatment and by 0.95% and 1.21% after 48 weeks of treatment. The co-administration of Dapa with pioglitazone resulted in less weight gain and less edema in comparison with pioglitazone monotherapy [33].

The effect of Dapa has been also examined when added on patients receiving sitagliptin with or without metformin. After 24 weeks of intervention, HbA1c levels were significantly reduced in the group treated with Dapa (by 0.5%) when compared with placebo [34]. The results were sustained through week 48.

Finally, a 24 week RCT randomized subjects with inadequately controlled T2DM receiving high doses of insulin with or without oral antidiabetic drugs to either Dapa 2.5, 5, 10 mg or placebo [35]. Dapa resulted in greater reductions in HbA1c when compared with placebo (0.4% with 2.5 mg, 0.49% with 5 mg and 0.57% with 10 mg) and these findings were maintained in the 24-week extension of the trial [35,36]. Interestingly, subjects who were treated with Dapa reduced their daily insulin dose by 0.63–1.95 units, whereas patients receiving placebo increased their daily dose by 5.65 units. Moreover, body weight was increased by 0.43 kg in those receiving placebo, while it decreased by 0.92–1.61 kg in patients receiving Dapa. Nevertheless, those allocated to Dapa reported more hypoglycemic events [35].

A meta-analysis that was published in 2014 and includes some of the aforementioned studies showed that when Dapa was added on conventional antidiabetic agents, HbA1c and FPG were reduced by 0.52% and 20 mg/dL, respectively [37]. Similarly, another meta-analysis reported that the administration of 10 mg of Dapa reduced HbA1c by 0.54% [38].

### 3.2. Canagliflozin (Cana)

Cana was the second commercially available SGLT2i after Dapa and several studies have been performed to examine its effect on glycemia. Firstly, Stenlof et al. randomized subjects with T2DM that were inadequately controlled with diet and exercise to receive Cana or placebo [39]. After 26 weeks of intervention, Cana 100 and 300 mg reduced HbA1c by 0.77% and 1.03%, respectively. As expected, no similar effect was reported with placebo [39]. FPG was reduced by 36 and 43 mg/dL with Cana 100 and 300 mg respectively, whereas 2 h PPG was reduced by 48 mg/dL and 65 mg/dL, respectively. The effect of Cana on HbA1c, FPG and 2 h PPG was sustained over the 52-week extension of the trial [40].

A meta-analysis of 6 RCTs assessed the efficacy of Cana when added in subjects who were treated with metformin monotherapy [41]. Administration of Cana 100 and 300 mg reduced HbA1c by 0.59% and 0.74%, respectively. Similarly, FPG was reduced by 27 mg/dL and by 32 mg/dL, respectively [41].

Regarding the glycemic effect of Cana when added in patients receiving metformin and SU, Wilding et al. showed that HbA1c was significantly reduced with Cana 100 and 300 mg versus placebo after 26 weeks of treatment (−0.85%, −1.06%, and −0.13% respectively, *p* < 0.001). These reductions were maintained at week 52 (−0.74%, −0.96%, and 0.01%, respectively) [42].

Furthermore, another RCT reported that the addition of Cana 100 or 300 mg in patients inadequately controlled with metformin and pioglitazone, led to significantly lower HbA1c when compared with placebo (−0.89% versus −1.03%, respectively) [43]. The reductions with Cana were maintained in the 52-week extension of the study (−0.92% and −1.03%, respectively) [43].

The effect of Cana on glycemia was also examined in people with T2DM who were treated with metformin and sitagliptin [44]. After 26 weeks of intervention, HbA1c was significantly reduced in the group treated with Cana 100 or 300 mg (pooled 0.91%) in comparison with placebo [44]. In addition, Cana resulted in significant reductions in FPG by 30 mg/dL [44].

Finally, when Cana was administered to patients with T2DM that were inadequately controlled with insulin, diet, and exercise, HbA1c was reduced by 0.97% after 16 weeks of intervention [45].

In terms of head-to-head trials, a RCT that included treatment-naive patients showed non-inferiority of Cana 100 and 300 mg, in terms of both HbA1c reduction and achievement of HbA1c < 7% in comparison with metformin [46]. Moreover, another study showed that among patients treated with metformin, the administration of Cana 300 mg led to similar glycemic improvements compared with glimepiride after 104 weeks of intervention [47]. Furthermore, a meta-analysis of three RCTs showed that Cana 300 mg significantly reduced HbA1c by 0.24% when compared with sitagliptin 100 mg [48]. Data from a real-world setting showed that among patients with T2DM, the addition of Cana resulted in a reduction of HbA1c by 1.16%, while the addition of GLP1-RA by 1.21% [49]. The definitions “real-world setting” and “real-world data” stand for information and data obtained from observational studies and registries and not from RCTs.

Regarding the hypoglycemic effect of Cana, a meta-analysis from two studies on treatment-naive subjects and one study on people receiving metformin monotherapy did not associate Cana 300 mg with an increased hypoglycemic risk when compared with placebo [48]. Nevertheless, hypoglycemic events were significantly higher when added on insulin or SU [48]. Similar results were reported from another meta-analysis, which showed that patients treated with Cana 100 or 300 mg experienced more hypoglycemic episodes than placebo [50].

### 3.3. Empagliflozin (Empa)

Empa is the third SGLT2i that is commercially available in Europe and in the USA. Several phase III RCTs have examined its glycemic efficacy. Firstly, Empa’s effect on glycemia was compared with placebo in untreated patients with T2DM [51]. After 24 weeks of intervention, HbA1c was reduced by 0.74% for Empa 10 mg and by 0.85% for Empa 25 mg [51]. Apart from placebo, there was a treatment arm allocated to sitagliptin, in which HbA1c was reduced by 0.73%. In a subgroup analysis of patients with HbA1c ≥ 8.5% at baseline, both doses of Empa led to greater HbA1c reductions than sitagliptin (−1.44% with Empa 10 mg, −1.43% with Empa 25 mg, −1.04% with sitagliptin). The authors of the study suggested that the extent of Empa’s effect depends partially on the degree of glycemia [51].

In addition, Haring and coworkers performed a study to investigate the efficacy and tolerability of Empa as an add-on to metformin monotherapy in patients with T2DM [52]. After 24 weeks of treatment, changes from baseline HbA1c were −0.13% with placebo, −0.70% with Empa 10 mg, and −0.77% with Empa 25 mg [52]. The trial included an arm treated with Empa 25 mg, for patients who had HbA1c > 10% at baseline. After 24 weeks of intervention, HbA1c was decreased from 11.1% to 7.9% (mean change from baseline −3.2%).

Another RCT assessed the glycemic effect of Empa in patients treated with metformin plus SU [53]. At week 24, HbA1c was reduced by 0.82% and 0.77% in patients treated with Empa 10 and 25 mg, respectively. Hypoglycemic events were reported more often in the group treated with Empa when compared with placebo [53]. This might suggest an increased risk of hypoglycemia with Empa when combined with SUs due to the known glucose-independent insulin-releasing effect of SUs.

Similar improvements in HbA1c were demonstrated in a 24-week RCT in which Empa was added on people with T2DM receiving pioglitazone with or without metformin [54]. The change in HbA1c was −0.6% and −0.7% with Empa 10 mg and 25 mg, respectively, versus −0.1% with placebo [54]. Edema, heart failure (HF) and bone fracture are common side effects associated with pioglitazone treatment. Interestingly, no increase in these adverse effects was observed in patients receiving Empa when compared with placebo.

In addition, another study evaluated the efficacy of Empa versus placebo when added in patients with inadequate glycemic control treated with linagliptin and metformin [55]. After 24 weeks, Empa 10 mg and 25 mg significantly improved HbA1c by 0.79% and 0.70% respectively, versus placebo. FPG was significantly reduced in both Empa groups [55]. Unexpectedly, reductions in mean HbA1c with the two different doses of Empa were similar in this trial.

A recently published trial examined the efficacy of Empa when added in subjects treated with liraglutide [56]. After 52 weeks of intervention, Empa 10 mg and 25 mg reduced HbA1c by 0.55% and 0.77% respectively, and FPG by 32.5 and 36 mg/dL, respectively [56].

Finally, Rosenstock and colleagues investigated the efficacy and tolerability of Empa when added in patients treated with basal insulin [57]. At week 18, HbA1c was reduced by 0.6% and 0.7% for Empa 10 mg and 25 mg, respectively [57].

In terms of head-to-head trials, one study compared Empa with glimepiride for a period of 2 years in people with T2DM that were inadequately controlled with metformin [58]. The administration of Empa resulted in an HbA1c reduction of 0.11% (*p* = 0.0153 for superiority) when compared with glimepiride. On top of that, fewer subjects experienced hypoglycemic events with Empa than with glimepiride administration.

### 3.4. Ertugliflozin (Ertu)

Ertu is the most recently approved SGLT2i for the treatment of T2DM. Firstly, a study by Terra et al. examined the effect of Ertu on glycemia when added in patients inadequately controlled with diet and exercise [59]. After 26 weeks of intervention, HbA1c was reduced from baseline by 0.99% and 1.16% for Ertu 5 mg and 15 mg doses, respectively [59]. Both FPG and 2 h PPG levels were significantly reduced with Ertu. The results were maintained in the 52-week extension of the study [60].

In addition, another RCT was performed to assess the efficacy of Ertu when administered in patients with T2DM that were treated with metformin [61]. After 26 weeks of intervention, the mean change from baseline HbA1c was −0.7% and −0.9% for Ertu 5 mg and 15 mg, respectively [61].

Another phase III study assessed the efficacy and safety of the combination of Ertu plus sitagliptin compared with placebo in patients with T2DM inadequately controlled with diet and exercise [62]. After 26 weeks, HbA1c was reduced by 0.4%, 1.6%, and 1.7% for the groups randomized to placebo, Ertu 5 mg plus sitagliptin 100 mg and Ertu 15 mg plus sitagliptin 100 mg, respectively. Furthermore, FPG and 2 h PPG were significantly reduced in both Ertu plus sitagliptin groups in comparison with placebo [62].

When Ertu was compared with glimepiride in terms of glycemic control in patients with T2DM inadequately controlled with metformin, the change in HbA1c was −0.6%, −0.6%, and −0.7% in the Ertu 15 mg, 5 mg, and glimepiride groups, respectively [63]. In addition, the incidence of symptomatic hypoglycemia was higher with glimepiride. Hence, Ertu is non-inferior to glimepiride in reducing HbA1c when added to metformin in patients with T2DM. These findings were confirmed in the extension of the study [64].

## 4. Cardiovascular Effect of SGLT2i

### 4.1. Cardiovascular Trials

In 2008 the FDA and in 2012 the EMA required CV safety trials for all new antidiabetic medications. These studies do not assess efficacy for glycemic control, but non-inferiority for CV outcomes of the new drugs, while superiority is a secondary outcome. Generally, these trials enroll subjects with high CV risk in order to gather a sufficient number of CV events in a short period of time [6].

The Empagliflozin Cardiovascular Outcome Event Trial in Type 2 Diabetes Mellitus Patients (EMPA-REG OUTCOME) examined the CV safety of Empa [65]. Briefly, 7020 patients with T2DM and coronary, peripheral or cerebrovascular disease were randomized to receive two different doses of Empa (10, 25 mg) or placebo for a median observation period of 3.1 years [65]. All participants were treated holistically in terms of CV protection and received lipid-lowering medication, antiplatelets and renin-angiotensin-aldosterone system inhibitors (RAASi). The study showed that the 3P-MACE (death from CV causes, non-fatal myocardial infarction (MI) or non-fatal stroke) primary composite outcome occurred in a significantly lower percentage (14%) of patients in the Empa group in comparison with the group treated with placebo. Regarding the secondary endpoints, treatment with Empa resulted in a 38% reduction of death from CV causes, 32% reduction of death from any cause and 35% reduction of hospitalization for HF, while no significant effect was observed in MI and stroke events. Intriguingly, there was a very early divergence of the CV events curves between Empa and placebo treatment groups. The number needed to treat (NNT) for Empa was 39, meaning that 39 patients would need to be treated during a 3-year period to prevent one death from CV causes [65]. Comparatively, the NNT for simvastatin is 30 and for ramipril is 56. While it should also be noted that the population in the latter studies did not receive high intense treatment for CV disease, such as statins or RAASi [66,67].

The CANVAS (Canagliflozin Cardiovascular Assessment Study) Program trial consisted of a combination of two sub-studies: the CANVAS and the CANVAS-R study (CANVAS-Renal). The former was designed to assess the CV safety of Cana, while the latter was conducted to investigate the effect of Cana on albuminuria [68]. With the merge of these two studies, the effect of two different doses of Cana (100 and 300 mg) versus placebo on CV disease was examined. A total of 10,142 patients with T2DM and either established CV disease or multiple CV risk factors were recruited. The mean follow-up was 3.6 years. Participants were patients treated well in the context of routine CV protective regimens using statins, antiplatelets and RAASi [68]. The study demonstrated that there was a significant reduction (by 14%) of the composite 3P-MACE primary endpoint in the group receiving Cana. In addition, Cana reduced the incidence of HF hospitalization, while no significant effect was reported in all-cause and CV mortality [68].

Moreover, the results of the Dapagliflozin Effect on Cardiovascular Events (DECLARE-TIMI 58) study, which is the CV trial of Dapa, have been recently published [69]. A total of 17,160 patients with T2DM and established atherosclerotic CV disease or multiple risk factors for atherosclerotic CV disease were randomized to receive either Dapa 10 mg or placebo for a median period of 4.2 years. The primary outcomes of the study were the 3P-MACE and a composite outcome of CV death or hospitalization for HF [69]. Treatment with Dapa resulted in a 17% reduction of the composite outcome of CV death or hospitalization for HF, while no effect was reported for the 3P-MACE. However, when each component of the outcome was analyzed individually, the effect was clearly driven by a strong reduction in hospitalization for HF, with no evidence of change in CV death [69]. In terms of secondary endpoints, Dapa reduced the composite renal outcome by 24% (≥40% decrease in eGFR to <60 mL/min/1.73 m^2^, new end-stage renal disease, or death from renal or CV causes), but had no effect on death from any cause [69].

The long-term effects of Ertu on CV and renal outcomes are being assessed in the Evaluation of Ertu efficacy and safety cardiovascular outcomes trial (VERTIS-CV) [70]. A total of 8238 patients with T2DM and established CV disease have been recruited. The primary outcome is the composite 3P-MACE and the key secondary outcomes are CV death or hospitalization for HF, CV death and renal death, dialysis/transplant, or doubling of serum creatinine from baseline [70]. The estimated study completion date is September 2019.

The basic characteristics of the CV safety studies of SGLT2i are presented in Table 1.

### 4.2. Meta-Analyses and Real-World Data

A recently published meta-analysis of the EMPA-REG OUTCOME, the CANVAS Program, and the DECLARE-TIMI 58 trials included a total of 34,322 patients with T2DM [71]. Among them, a total of 20,650 (60%) people had established CV disease and 3891 (11.3%) had HF. SGLT2i reduced major cardiac events (MI, stroke, or CV death) significantly by 11% [71]. This effect was present only in patients with atherosclerotic CV disease, while no such effect was observed in subjects without established CV disease. On the other hand, SGLT2i reduced the risk of CV death or hospitalization for HF by 23% with a similar benefit in patients with and without atherosclerotic CV disease as well as with and without a history of HF at baseline. Furthermore, SGLT2i significantly reduced the risk of MI by 11% and CV mortality by 16%, whereas SGLT2i had no effect on strokes [71]. In addition, SGLT2i reduced significantly hospitalization for HF by 31%. Regarding all-cause mortality, SGLT2i reduced the risk by 15%, however, the heterogeneity was high [71]. The CV effects of SGLT2i apply also to individuals with T2DM and CKD [72]. More specifically, a recent meta-analysis which included patients with T2DM and eGFR < 60 mL/min/1.73 m^2^ showed that intervention with SGLT2i reduced significantly the risk of CV death, non-fatal MI or non-fatal stroke and HF, without a clear effect on all-cause mortality [72].

Many scientists criticize RCTs since they do not represent every day clinical practice. Therefore, real-world data were published to clarify whether the cardioprotective effects of SGLT2i exist in a real-world clinical setting. Birkeland et al. performed an observational analysis of data from Denmark, Norway and Sweden [73]. A total of 22,830 patients with T2DM received SGLT2i (mainly Dapa) and 68,490 were treated with other glucose-lowering agents for a mean follow-up of 0.9 years (80,669 patient-years). About 25% of them had established CV disease at baseline. The administration of SGLT2i was associated with a significantly decreased risk of CV mortality, major adverse CV events and hospitalization for HF, when compared with other glucose-lowering drugs. No significant differences were reported between the use of SGLT2i and the use of other glucose-lowering drugs regarding non-fatal MI or non-fatal stroke [73]. Interestingly, the difference in CV mortality was similar for the patients with established CV disease, while for major CV events the change was significant only for those with CV disease at baseline. Therefore, these data extend the results of the CV RCTs to a real-world clinical setting in an unselected population of patients with T2DM and a broad CV risk profile [73]. Another real-world study from the same group included 309,056 patients newly initiated on either SGLT2i or other glucose-lowering drugs [74]. Among them, a total of 13% had established CV disease at baseline. Treatment with SGLT2i was associated with significantly lower rates of hospitalization for HF, all-cause death, and hospitalization HF or all-cause death [74]. It should be emphasized that since only a small percentage of people have established CV disease in real-world studies, the CV benefits of SGLT2i might be applicable to people with or without CV disease.

### 4.3. SGLT2i and Traditional Cardiovascular Risk Factors

#### 4.3.1. Body Weight

Obesity is an independent risk factor for CV events [75]. SGLT2 inhibition leads to increased glucose excretion in the urine. It is estimated that 75 gr glucose per day (approximately 300 kcal/day) are lost in the urine with a diuresis of 400 mL/day [76]. Data from clinical trials so far show that the total weight loss seen by these drugs is 2–3 kg [76]. Weight loss is seen from the first weeks of treatment, reaches a plateau after 6 months and is maintained for a long time [16,76]. However, the expected energy loss is not translated to expected weight loss since the anticipated weight reduction would have been greater [76]. SGLT2i do not have effect on resting or postprandial energy expenditure, hence, the difference between expected weight loss and observed weight loss implies gradual compensatory increase in caloric intake [76,77].

Among SGLT2i, a meta-analysis showed that Cana 300 mg leads to greater weight reduction when compared with Dapa 5 mg (−0.89 kg), whereas Cana 100 mg does not seem to differ from other SGLT2i [78].

#### 4.3.2. Blood Pressure

It is widely known that the reduction of arterial blood pressure (BP) is associated with reduction of CV morbidity and mortality in patients with DM [79]. In the EMPAREG-OUTCOME trial, Empa managed to reduce both systolic and diastolic BP without increasing heart rate [65]. These results were reproduced in several studies, and two meta-analyses have established the beneficial effect of SGLT2i on BP. More specifically, SGLT2i reduce systolic BP by 2.46 mmHg and diastolic BP by 1.46 mmHg, while they also reduce 24-h ambulatory systolic and diastolic BP by 3.76 mmHg and 1.83 mmHg, respectively [80,81]. Several underlying pathophysiologic mechanisms have been proposed for this effect. Firstly, the inhibition of the co-transporters in the proximal tubule leads to a mild increase of sodium urine excretion, while increased glucose excretion per se causes an additional osmotic diuretic effect. Secondly, weight loss and reduction of sympathetic nervous activity have been implicated in the reduction of BP [11]. In addition, beneficial effects of SGLT2i on arterial stiffness (as discussed below) may affect the BP.

Even though all SGLT2i reduce BP, indirect data from a meta-analysis demonstrated that Cana 300 mg led to greater reduction of systolic BP in comparison with other SGLT2i, while no differences were reported for diastolic BP among several SGLT2i [50].

Nevertheless, even BP lowering drugs have not been so effective and so rapid in reducing CV events and especially HF as SGLT2i has [82].

#### 4.3.3. Renal Outcomes

CKD increases CV risk in people with and without DM [83]. All three CV trials of Empa, Cana, and Dapa showed impressive results in renal outcomes [65,68,69]. Sub-analysis from the EMPA-REG OUTCOME trial showed that in people with established CV disease, treatment with Empa resulted in reduced incident or worsening of nephropathy when compared with placebo [84]. This reduction was seen in several subgroups of the population and especially in those with CKD [84]. However, it should be noted that in this study renal events were recorded from investigators and were not adjudicated.

Regarding Cana, the Evaluation of the Effects of Canagliflozin on Renal and Cardiovascular Outcomes in Participants with Diabetic Nephropathy (CREDENCE) trial was performed in subjects with T2DM and nephropathy to examine the effect of Cana on renal function [85]. In contrast to the EMPA-REG OUTCOME, this trial had a composite primary endpoint that consisted of renal events which were adjudicated. Administration of Cana led to the reduction of end-stage renal disease, doubling of the serum creatinine level, or death from renal or CV causes [85]. The Study to Evaluate the Effect of Dapagliflozin on Renal Outcomes and Cardiovascular Mortality in Patients With Chronic Kidney Disease (Dapa-CKD) examines the effect of Dapa on renal function in people with CKD and is estimated to be completed in November 2020 [86].

SGLT2i inhibit the reabsorption of sodium and glucose from the tubule and hence, more sodium is delivered in the macula densa causing afferent arteriole dilation, reduced intraglomerular pressure and decreased hyperfiltration [87]. This is presented clinically with a decline in the eGFR by 4–5 mL/min/1.73 m^2^ during the first weeks of treatment and a return to baseline values after 6–12 months [10,16]. Approximately 7% of plasma volume is reduced due to diuresis which leads to contraction in plasma volume [88]. Interestingly, this effect is maintained during SGLT2i therapy, in contrast to treatment with thiazides [87]. In addition, the improvement of traditional CV factors such as arterial BP and weight, as discussed before, may also have a favorable effect on renal function over time. Nevertheless, it is unlikely that the latter can fully explain the impressive effect of SGLT2i on renal outcomes since patients in these studies were treated with RAASi [87]. Therefore, other mechanisms have been proposed.

Firstly, it is known that SGLT2i cause natriuresis and volume depletion and hence, circulating levels of renin, angiotensin and aldosterone increase [89]. When SGLT2i are co-administered with RAASi they activate the Mas receptor that leads to systemic arteriolar vasodilation, natriuresis, reduced oxidative stress and anti-proliferative activity by increasing the production of nitric oxide and prostaglandin [87]. Secondly, SGLT2i increase kidney oxygen consumption by using ketones instead of free fatty acid [87]. Furthermore, other mechanisms such as increased erythropoietin and thereby hematocrit, due to decreased renal cortical oxygen tension or the interaction between SGLT2 and the sodium–hydrogen exchanger (NHE), have been proposed [89].

Consequently, the improvement in kidney function may contribute to the decreased CV events in people with DM. However, it should be noted that the mechanisms explaining the beneficial effects of SGLT2i on kidney function are beyond the purpose of this article.

#### 4.3.4. Lipid Profile

Dyslipidemia is a common comorbidity of T2DM that increases CV morbidity and mortality [90]. The EMPA-REG OUTCOME trial and the CANVAS program showed that the administration of Empa or Cana increased both low-density lipoprotein cholesterol (LDL-C) and high-density lipoprotein cholesterol (HDL-C) when compared with placebo [65,68]. These results were confirmed from a meta-analysis of Empa trials that demonstrated an increase of LDL-C in the group treated with Empa [91]. In addition, a meta-analysis of 34 RCTs showed that the administration of SGLT2i (Dapa, Cana, Empa) increased HDL-C (mean difference 1.93 mg/dL), LDL-C (mean difference 3.5 mg/dL) and decreased serum triglycerides (mean difference 7.8 mg/dL [92]. Cana was associated with the largest effects on serum lipids [92]. Another recent meta-analysis demonstrated similar results [93].

Except for the quantitative effects of SGLT2i on serum lipids, studies with SGLT2i have shown changes in lipoprotein sub-fractions [94]. A Japanese study reported that Dapa administration suppressed atherogenic small dense LDL-C, whereas it increased HDL2-C, which is a favorable cardiometabolic marker [95]. In addition, treatment with Dapa increased the levels of the less atherogenic large buoyant LDL-C [95].

To sum up, SGLT2i administration is associated with a small increase in LDL-C and HDL-C levels, while triglyceride and small dense LDL levels tend to modestly decrease [94].

#### 4.3.5. Arterial Stiffness and Endothelial Function

Pulse wave velocity is the “gold-standard” method for the assessment of arterial stiffness and independently predicts CV disease morbidity and mortality [96]. One study showed that pulse wave velocity was reduced after 48-h administration of Dapa in 16 patients with T2DM (10.1 ± 1.6 to 8.9 ± 1.6 m/s, *p* < 0.05) [97]. However, no safe conclusion can be drawn from one study [98]. In addition, the same study by Solini et al. demonstrated the beneficial effect of Dapa’s administration on endothelial function as assessed with the flow-mediated dilation method [97]. Authors suggested that Dapa might have a direct effect on the vascular endothelium since known parameters that cause endothelial dysfunction, such as hyperglycemia and sympathetic activation, were not found to affect endothelial function [97]. In addition, another study with a duration of 6 months showed that the administration of Dapa increased the reactive hyperemia index, which is another marker of endothelial function [99]. However, more studies are needed to determine whether this is a class effect. An ongoing study is examining the effect of Empa on endothelium in patients with T2DM and established CV disease [100]. Apart from human studies, in vitro studies have reported that SGLT2i have a beneficial effect on endothelial function [101].

### 4.4. Established Cardioprotective Mechanisms

The aforementioned effects of SGLT2i on glycemic profile, body weight, lipid panel, BP, endothelial function and arterial stiffness are unlikely to fully account for the beneficial CV outcomes of SGLT2i. Even though they do have an impact on CV disease, it would take years for the reduction of mortality to become apparent. Lately, hundreds of manuscripts have been published with the aim of explaining the pathophysiological mechanisms behind the CV benefits of SGLT2i. The leading hypotheses include improvement in cardiac metabolism and bioenergetics, improvement in ventricular loading conditions, inhibition of the Na^+^/H^+^ exchange, alteration in adipokines and cytokines production, as well as the reduction of necrosis and cardiac fibrosis [102,103,104].

#### 4.4.1. SGLTi and Metabolism Alteration

Mitochondrial oxidative metabolism is responsible for most (95%) of the production of myocardial energy, and the normal fuel for this process is balanced among free fatty acids (FFA) (70%), glucose (20%), and, to a lesser degree, lactate, amino acids and ketone bodies [102,104]. A healthy heart is able to rapidly switch from one source to another depending on substrate availability, workload, hormonal milieu and level of tissue perfusion [102]. In a diabetic heart, this metabolic flexibility is impaired and myocardium becomes more dependent on FFA oxidation as energy fuel [104,105]. The increased utilization of this substrate as an adenosine triphosphate (ATP) generator may lead to overproduction of FFA intermediates, which may promote lipotoxicity and impairment in sarcoplasmatic reticulum, further resulting in ventricular stiffness and dysfunction [102,103,106].

It is hypothesized that SGLT2i lead to a starvation situation, which ultimately increases FFA and ketone body levels. Several mechanisms may contribute to an increase in ketone bodies. The glucose lowering effect of these drugs results in a decrease in endogenous insulin secretion from β-cells as well as in the doses of exogenous administered insulin [107,108]. On the other hand, the inhibition of SGLT2 receptors on α-cells seems to enhance glucagon production, by lowering intracellular glucose levels [109]. The net result is an increase in glucagon to insulin ratio, which promotes lipolysis and ketogenesis [107]. Apart from an increase in production, SGLT2i seem to deteriorate the ketone bodies’ excretion from kidneys by reducing glomerular filtration, simultaneously with a stimulation of tubular reabsorption [108].

Thereby, SGLT2 inhibition offers an alternative fuel in diabetic myocardium, the b-hydroxybutyrate, which competes FFA’s and glucose’s entry in myocardial mitochondrial metabolic oxidation [110,111,112]. The cardioprotective effects of this substrate are several-fold: it maintains the mitochondrial integrity by generating less reactive oxygen species (ROS) production, it stabilizes cell membrane potential offering an anti-arrhythmic effect and it may prevent pro-hypertrophic transcription pathways by inhibiting histone deacetylase [113,114,115]. It is also postulated that SGLT2i stimulate branched-chain amino acid (BCAA) degradation, which is impaired in HF, offering an alternative fuel source in failing diabetic myocardium [116].

In addition, in vitro studies using murine and human ventricular cardiomyocytes demonstrated that Empa increases the glucose transporter 1 expression and enhances glucose uptake [117].

Although the findings described above are tempting, further investigation should be addressed to elucidate the beneficial effects of SGLT2 inhibition on cardiac metabolism and bioenergetics.

#### 4.4.2. SGLTi and Ventricular Loading Conditions

As mentioned before, inhibition of the SGLT2 in the proximal tubule leads to natriuresis apart from glycosuria. This osmotic diuresis has a positive impact in Franklin–Starling curve of left ventricle in patients with T2DM due to reduction in preload volume [118,119]. As aforementioned, several studies have showed that this new anti-hyperglycemic class leads to reduction in BP, arterial stiffness and ventricular resistance, effects that could improve the cardiac function of patients with DM by acting on the second most vulnerable factor of cardiac output, the afterload.

#### 4.4.3. SGLTi and Inhibition of the Na^+^/H^+^ Exchange

The NHE isoform 1 is expressed in the myocardium and a tempting hypothesis is that SGLT2i may directly inhibit this exchanger, which is upregulated in patients with HF [120,121]. The NHE 1 is of particular importance in the myocardium because it has been associated with abnormal myocardial hypertrophy and heart ischemia-reperfusion injury, due to increase in the intracellular calcium and sodium concentration [122]. The cardioprotective effect of SGLT2i through this mechanism was established in an experimental study, in which the use of Empa led to reduction of myocardial cytoplasmatic calcium and sodium, while mitochondrial calcium levels were increased [123]. In addition, SGLT2i seem to downregulate another isoform of this exchanger in kidneys, the NHE 3, which is increased in HF patients, and thus these agents seem to promote natriuresis [120,124].

#### 4.4.4. SGLTi Inhibition and Adipokines

Adipokines released by epicardial and perivascular adipose tissue have been implicated in the outset of HF, through endocrine and paracrine effects. It has been demonstrated that some adipokines such as leptin, promote inflammation of the myocardium, while others such as adiponectin, have anti-inflammatory and cardioprotective effects. SGLT2i have been proposed as a means of restoring the equilibrium between these pro- and anti-inflammatory adipokines, thus eliminating myocardial dysfunction. A recent study by Garvey et al. showed that Cana led to a 25% reduction of serum leptin levels and to a 17% increase of serum adiponectin when compared with glimepiride [125]. Another study by Sato et al. showed that Dapa reduces ectopic epicardial fat that plays a crucial role in the genesis of HF [126]. Even though various mechanisms have been suggested, it is not elucidated in which way SGLT2i alter adipokine levels, because it is not known if SGLT2i actions are primarily exerted by adipose tissue function or from fat loss. Recent research showed that phloretin (a breakdown product of phlorizin) directly inhibited leptin secretion in adipocytes in a dose-dependent manner [127].

#### 4.4.5. SGLTi and Cardiac Fibrosis

Myocardial fibrosis is an essential part of cardiac remodeling that leads to HF. Aberrantly activated fibroblasts secrete extracellular matrix proteins in myocardium, leading to altered ventricular performance and contractile dysfunction [128]. Lee et al. showed that the administration of Dapa has a significant cardiac anti-fibrotic effect in rat models of post-myocardial infarction, by reducing collagen synthesis via stimulating M2 macrophages and by inhibiting myofibroblast differentiation [129]. In addition, Kang et al. provided data that Empa suppresses pro-fribrotic markers such as type I collagen, a-smooth muscle actin, connective tissue growth factor and matrix metalloproteinase 2 and attenuates TFG-β1-induced fibroblasts activation [130].

It is obvious that SGLT2i have beneficial effects on one of the most critical factors of HF, the cardiac fibroblasts.

## 5. Adverse Effects of SGLT2i

An observation based on case reports that treatment with SGLT2i may be associated with an increased risk of diabetic ketoacidosis (DKA) led to a warning from the FDA in May 2015 [131] as well as from the EMA in June 2015 [132].

As discussed previously, SGLT2i lead to increased circulating ketone bodies. Several factors such as profound insulin deficiency, which is more common in type 1 DM, low carbohydrate diet, dehydration, excess alcohol intake, major illness or post-surgery period can trigger the development of severe DKA [133]. DKA related to SGLT2i may be accompanied by low glucose levels (<250 mg/dL), due to the glycosuric effect of these drugs, and hence, it is called euglycemic [134]. The risk of DKA with the SGLT2i use varies from study to study. The incidence of DKA was very low and non-significant in both CANVAS and EMPA-REG OUTCOME trials, while in DECLARE-TIMI 58 a small but statistically significant (*p* = 0.020) number of events was observed [65,68,69]. A possible explanation for this lack of agreement could be the fact that DECLARE-TIMI 58 was the most recent trial and the awareness of DKA with SGLT2i was high. A recent meta-analysis of RCTs by Monami et al., however, reported a non-significant increase (*p* = 0.780) in the prevalence of DKA with SGLT2i compared with placebo or other antidiabetic drugs [135], and similar were the results of another meta-analysis by Wang et al. [136]. Real-world data, on the other hand, indicate an increased risk for DKA among SGLT2i users. According to an analysis of the FDA adverse effects reporting system (FAERS) by Fadini et al., the proportional reporting ratio (PRR) of DKA in reports including versus those not including an SGLT2i was 7.9 (95% Confidence Intervals (CI): 7.5, 8.4) for the whole population [137]. When SGLT2i were administered off-label to patients with type 1 DM, the relative risk for DKA rose up to 57.3 [137]. A predominance of female gender among DKA cases was also observed [137]. Likewise, Fralick et al. used the records of 140,352 patients in the United States and observed a higher incidence of DKA (Hazard Ratio (HR) 2.2, 95% CI 1.4–3.6) among patients recently introduced in SGLT2i treatment compared with those who were recently prescribed a DPP4i [138]. Comparable were the results of another study, which, using the FAERS data, reported a 7-fold risk for DKA among patients with T2DM on treatment with SGLT2i versus those receiving DPP4i [139]. The lower incidence of DKA in RCTs compared with pharmacovigilance data analysis may be explained by the presence of controlled conditions and cautious selection of participants in RCTs.

The higher urine glucose levels due to the glycosuric effect of SGLT2i, predisposes patients to urinary tract and genital infections [140]. Available studies, however, provide conflicting data about the incidence of urinary tract infections (UTIs) in patients on treatment with SGLT2i. In the three large CV outcome trials, no significant increase in UTIs was observed with each drug when compared with placebo [65,68,69]. A 2013 meta-analysis by Vasilakou et al. reported a higher risk for UTI with SGLT2i [141], while two subsequent meta-analyses failed to show any significant difference in UTIs between SGLT2i and either placebo or active comparators [142,143]. In drug specific meta-analyses however, Dapa alone was associated with a significantly higher risk for UTIs [142,143]. Similar were the results drawn from real-world data. The frequency of UTIs was not significantly increased with SGLT2i versus either DPP4i or GLP1-RA in two studies [144,145]. In contrast, genital infections seem to be consistently more frequent in patients treated with SGLT2i. A significant increase in genital infections was observed in the EMPA-REG OUTCOME as well as in the CANVAS and DECLARE-TIMI 58 trial [65,68,69]. Meta-analysis of RCTs confirm an increase in the risk of genital infections with SGLT2i versus either placebo or an active comparator [48,91,141,142,143]. Studies based on analysis of post-marketing data led to similar conclusions [144,146,147]. Female gender and previous history of genital infection were found to increase the chance for the infection to occur [148]. The genital infections, however, tend to be non-severe and manageable without the need for treatment discontinuation [149,150]. An exception consisted of Fournier’s gangrene, a rare but life-threatening condition. Only case reports indicate an association between SGLT2i treatment and Fournier’s gangrene occurrence. Nevertheless, due to the severity of the disease, the FDA released a warning [151].

It has been proposed that SGLT2i may alter mineral metabolism and thereby affect bone density and increase the risk for fractures [152,153]. Serum phosphate, magnesium and parathyroid hormone may be increased with SGLT2i treatment, while only modest changes in serum or urinary calcium concentrations and vitamin D have been reported [152,153]. A significant increase in serum collagen type 1 beta-carboxy telopeptide (beta-CTX), a bone resorption marker, as well as in serum osteocalcin, a bone formation marker, has also been observed with Cana [154]. Decreases in total hip bone mineral density (BMD) have been noted as well after two years of Cana treatment, most likely mediated by a decrease in body weight, while the rest of the skeleton seems to remain uninfected [154]. The risk of fractures overall, but not of low-trauma fractures, was significantly higher with Cana versus placebo, in the CANVAS trial, while a non-significant difference was observed with Empa and Dapa in the two large CV outcome studies [65,68]. Additionally, three recent meta-analysis did not confirm an increased incidence of bone fractures with SGLT2i versus either placebo or active treatment [155,156,157]. In a population-based cohort study, Cana was not related to increased incidence of fractures, compared with GLP1-RA [158], and similar were the results of another nationwide study that included all SGLT2i available in Europe [145]. An increased risk for falls due to orthostatic hypotension or a direct effect on bone metabolism could be the underlying mechanism for the increased incidence of fractures in some of the aforementioned studies. Overall, however, current data are rather reassuring about the risk of fractures with this class of drugs.

An almost double incidence of lower limb amputations (LLA) was observed with Cana versus placebo (6.3 versus 3.4 participants with amputation per 1000 patient-years, HR: 1.97, 95% CI: 1.41–2.75), while non-significant was the difference observed with Empa in the EMPA-REG OUTCOME TRIAL and with Dapa in the DECLARE-TIMI trial [65,68,69]. As a consequence, a relevant warning was released from the FDA for Cana alone, while the EMA included all the three drugs available in Europe [159,160]. Differences in the design of the three studies and the collection of the data regarding LLA may partly explain the differences in the results. In a recent meta-analysis of 14 RCTs by Li et al., likewise, no increase in LLA was reported with SGLT2i as a class. In subgroup analysis, however, a significantly higher risk for LLA was reported with Cana versus placebo or non-SGLT2i antidiabetic drugs, but not with Empa [161]. Comparable were the results of two other meta-analyses. The study by Kohler et al., which included RCTs with Empa alone, found no difference in the incidence of amputations among the two treatment groups and the study by Jabbour et al. reported no increase in LLA with Dapa versus placebo or active comparators [162,163]. The pathophysiological mechanism that may mediate the increase in amputations observed with Cana is currently unknown. It has been proposed that volume depletion induced by SGLT2i may worsen perfusion of already dysfunctional vascular network, but this hypothesis has not been proven yet [164]. The findings of observational cohort studies, on the other hand, were heterogeneous. In one study, a lower risk for LLA was reported with SGLT2i when compared with SUs, accompanied by a similar trend when compared with DPP-4i [165]. Opposite to this, the results of another study were non-statistically significant [166]. In a pharmacovigilance analysis of FAERS data, an increased risk of LLA was observed with Cana, but not with Empa or Dapa [167]. Additionally, in a cohort study that included registers from Sweden and Denmark, the incidence of amputations was also higher with SGLT2i versus GLP1-RA [145]. In two other studies based on real-word data, however, no significant increase in LLA was observed with SGLT2i [168,169]. Limited data are available about the risk of amputations with the rest three SGLT2i (ipragliflozin, tofogliflozin, Ertu), these inhibitors do not seem to be associated with amputations [62,170,171]. Interestingly, Empa is recommended as second line agent after metformin for glycemic control in patients with diabetic foot ulcers and peripheral arterial disease [172]. Overall, an increased risk for LLA has been observed with Cana, while other SGLT2i have not been consistently correlated with amputations. The exact mechanism that mediates this adverse event of Cana, if so, is not known for now and whether this is a class effect or not has to be clarified.

A concern about an increase in female breast cancer and in male bladder cancer with SGLT2i has been raised, based on trials with Dapa [141]. Results of DECLARE-TIMI, nevertheless, did not confirm a higher cancer risk with Dapa [69]. An early diagnosis rather than a true increase in cases of cancer may be the reason for the aforementioned observation. In addition, no increase in cancer prevalence was reported with either Cana or Empa [162,173]. One meta-analysis did not demonstrate any significant increase in cancer incidence with SGLT2i [174]. Of interest, a systematic review and meta-analysis observed a non-significant difference in the risk of overall cancer with SGLT2i when compared with placebo or active comparators, however, subgroup analysis revealed an increase in bladder cancer with SGLT2i in general and with Empa in particular, while a reduction in gastrointestinal cancers with Cana was also reported [175]. A possible explanation for an increased tumor genesis with SGLT2i, from bladder epithelium at least, could be the promotion of tumor growth by persistent glycosuria, in combination with chronic or recurrent urinary tract infections [176]. Current data, nevertheless, support that SGLT2i are rather safe and further research is needed to elucidate whether an increased risk for malignances is real or not.

A mild volume depletion may be provoked by SGLT2i, due to their osmotic diuretic effect. A slight reduction in BP, orthostatic hypotension and dizziness seem to be frequent with these agents, especially when combined with diuretics [174,177]. Nevertheless, one meta-analysis comparing Cana or Dapa with either placebo or active comparator and one comparing Empa with placebo, did not confirm a significant volume depletion with SGLT2i [91,178].

The risk of hypoglycemia with SGLT2i seems to be low. In the three large CV outcome trials the incidence of hypoglycemia was similar in the comparing groups [65,68,69]. In a meta-analysis, the risk of hypoglycemia was not significantly increased with SGLT2i versus the control group [174]. The incidence of hypoglycemia with SGLT2i overall tends to be similar to that with placebo or non- SGLT2i antidiabetic drugs, such as metformin, glitazones and DDP4i, and lower when compared with insulin or SUs [179]. An increased risk for hypoglycemia was however observed, when Empa, Cana or Dapa were added on treatment with insulin or SU [177]. A reduction in doses of insulin or SUs may be necessary when co-administered with a SGLT2i to prevent hypoglycemia.

In 2016 the FDA issued a warning of an increased risk of acute kidney injury (AKI) with Dapa and Cana, based on 101 confirmed cases of AKI reported to the FAERS [180]. Volume depletion due to osmotic diuresis, decrease in trans-glomerular pressure, as well as renal medullar hypoxic injury have been proposed as possible mechanisms that mediate renal damage by SGLT2i [181]. The incidence of AKI was not however higher with SGLT2i in the three CV outcome trials, CANVAS, EMPA-REG OUTCOME and DECLARE- TIMI [65,68,69]. In a meta-analysis on the other hand, both Cana and Dapa were associated with an increased risk for the composite renal outcome, while there was a non-significant trend towards an increased incidence of AKI [182]. Of interest, a significantly lower risk for AKI was observed with Empa compared with control group [182]. Studies based on real-world evidence are rather reassuring. Nadkarni et al. used data from the Mount Sinai CKD registry and the Geisinger Health System cohort and observed a lower risk for AKI among SGLT2i users versus non-users patients with DM [183]. In a nationwide register-based cohort study, likewise, the incidence of AKI did not differ significantly between SGLT2i and GLP1-RA users [145]. In conclusion, a higher risk for renal failure with SGLT2i is for now controversial. A careful selection of patients initiated on SGLT2i, however, as well as a close monitoring of eGFR, would be useful on behalf of health professionals.

## 6. Conclusions and Future Perspectives

DM is associated with increased CV risk and SGLT2i were the first antidiabetic drugs that demonstrated impressive CV benefits apart from their glucose-lowering effect. In this review, we have outlined the main effect of four commercially available SGLT2i on glycemic control and on CV disease. SGLT2i reduce HbA1c by 0.5%–1.0% by inhibiting the absorption of glucose from the proximal tubule of the kidney, causing glycosuria. They have additionally shown favorable effects on traditional CV factors such as body weight, BP, lipid profile, arterial stiffness and endothelial function. The predominant pathophysiological mechanisms that may explain the CV benefits of SGLT2i include plasma volume and diuresis, cardiac fibrosis, myocardial metabolism, as well as adipokine kinetics. Urinary tract and genital infections as well as euglycemic DKA are the most common adverse effects of SGLT2i. Other possible adverse effects include LLA, Fournier gangrene, bone fractures, female breast cancer, male bladder cancer, orthostatic hypotension and AKI.

Even though SGLT2i have revolutionized the treatment of T2DM, many questions remain unanswered. First, the exact pathophysiological mechanisms that can explain the CV and renal benefits of SGLT2i are not fully elucidated. Second, it is still unclear whether these effects can be generalized to the general diabetic population or are restricted to specific groups with cardiac or renal disease. Moreover, the beneficial effects of these drugs on cardiac and renal outcomes have been demonstrated mainly in subjects with HbA1c above 7% and it is not known if this also stands for individuals with HbA1c values <7%. Third, we are expecting the results of the studies that examine whether the favorable effects of SGLT2i on HF and renal function are reproduced irrespective of the presence of T2DM, as discussed before. Forth, among the SGLT2i that have completed several phase III and phase IV RCTs, all of them have in common the reduction of hospitalization for HF and the renal protective effects. However, in terms of CV death, all-cause mortality, as well as adverse effects, certain differences exist between them. Indicatively, Empa reduced significantly CV death, while Cana and Dapa did not. Similarly, Cana increased the incidence of LLA, whereas Empa and Dapa did not. Since the population of these studies was different it would be unfair to directly compare these agents. Thereby, head-to-head RCTs and well-designed observational studies as well as real-world data are needed to clarify whether these outcomes reflect a class-effect or there is indeed a difference between them. Then physicians who prescribe antidiabetic drugs would know better which one to use in every day clinical practice. In addition, the increase of the dose of SGLT2i does not seem to have major impact on glycemic control or renal function and does not increase the rate of adverse effects. However, the prescribing physician should follow the summary of products characteristics, the guidelines of scientific associations and local treatment protocols.

Finally, some authors suggest that since SGLT2i have shown such impressive benefits in CV morbidity and mortality but also in other metabolic disorders that often coexists with DM, such as excess body weight and arterial hypertension, they would substitute metformin as the first line single treatment in people with T2DM. However, head-to-head RCTs comparing SGLT2i with metformin are needed to clarify whether SGLT2i can safely replace metformin as a first-line agent in the management of DM.

## Figures and Tables

**Table 1 ijerph-16-02965-t001:** Basic characteristics of the cardiovascular safety studies of SGLT2i.

	EMPA-REG OUTCOME	CANVAS Program	DECLARE-TIMI 58	VERTIS-CV (Ongoing)
Intervention	Empagliflozin 10, 25 mg versus Placebo	Canagliflozin 100, 300 mg versus Placebo	Dapagliflozin 10 mg versus Placebo	Ertugliflozin 5, 15 mg versus Placebo
Population (*n*)	7020 patients with T2DM and established CV disease	10,142 patients with T2DM and established CV disease or ≥2 CV risk factors	17,160 patients with T2DM and established CV disease or risk factors for atherosclerotic CV disease	8246 patients with T2DM and established CV disease
Established CV disease (%)	99	66	41	99
Follow-up period (years)	3.1	3.6	4.2	-
HbA1c (%)	7.0%–10.0% on stable background therapy or 7.0%–9.0% for drug-naive patients	7.0%–10.5%	6.5%–12.0%	7.0%–10.5%
eGFR (mL/min/1.73 m^2^)	≥30	≥30	≥60	≥30
Primary outcome(s) (HR (95% CI))	3P-MACE 0.86 (0.74–0.99)	3P-MACE 0.86 (0.75–0.97)	- 3P-MACE 0.93 (0.84–1.03) - CV death or hospitalization for HF 0.83 (0.73–0.95)	3P-MACE -
Key Secondary outcome(s) (HR (95% CI))	4P-MACE 0.89 (0.78–1.01)	- All-cause mortality (below) - CV death (below) - Progression of albuminuria 0.73 (0.67–0.79) - CV death or hospitalization for HF 0.78 (0.67–0.91)	- ≥40% decrease in eGFR to <60 mL/min/1.73 m^2^ or new end-stage renal disease or death from renal/CV cause 0.76 (0.67–0.87), - All-cause mortality (below)	- CV death or hospitalization for HF - CV death - Renal death or dialysis/transplant or doubling of serum creatinine from baseline
Other Secondary Outcomes				
CV death (HR (95% CI))	0.62 (0.49–0.77)	0.87 (0.72–1.06)	0.98 (0.82–1.17)	-
All-cause mortality (HR (95% CI))	0.68 (0.57–0.82)	0.87 (0.74–1.01)	0.93 (0.82–1.04)	-
Fatal or non-fatal myocardial Infarction (HR (95% CI))	0.87 (0.70–1.09)	0.89 (0.73–1.09)	0.89 (0.77−1.01)	-
Fatal or non-fatal stroke (HR (95% CI))	1.18 (0.89–1.56)	0.87 (0.69–1.09)	1.01 (0.84–1.21)	-
Hospitalization for HF (HR (95% CI))	0.65 (0.50–0.85)	0.67 (0.52–0.87)	0.73 (0.61–0.88)	-

T2DM: type 2 diabetes mellitus; CV: cardiovascular; HbA1c: glycated hemoglobin; eGFR: estimated glomerular filtration rate; HF: heart failure; 3P-MACE: cardiovascular death, non-fatal myocardial infarction, non-fatal stroke; HF: heart failure; HR: hazard ratio; CI: confidence intervals; 4P-MACE: Cardiovascular death, non-fatal myocardial infarction, non-fatal stroke, hospitalization for unstable angina.

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
