# Peer review of "SGLT2 Inhibitors: A Review of Their Antidiabetic and Cardioprotective Effects"

_ijerph, 2019, doi:10.3390/ijerph16162965_

Round 1

Reviewer 1 Report

This a comprehensive and well referenced review on the effects of SGLT2 inhibitors. I have just minor points to make regarding style and/or grammar:

Please change the following sentences or amend as indicated:

"since it has been introduced 130 years ago by Belgian and French scientists": "since it was introduced 130 years ago by Belgian and French scientists"

"is referred as “splay”": "is referred to as “splay”" 

"is mediated mainly due to increased": "is mainly due to increased" 

"and it seems that is contributes": "and it seems to contribute"

"Dapa was approved from the EMA..." : "Dapa was approved by the EMA and by the FDA in 2014, Cana was approved by the FDA in 2013, while Empa was 93 approved by both EMA and FDA in 2014" 

"After 24-week of intervention": "After 24 weeks of intervention" 

"and the results were similar with those treated in the morning": "and the results were similar to those obtained by morning treatment"

RCT: please define acronym

"conventional antidiabetic agents HbA1c and FPG": "conventional antidiabetic agents, HbA1c and FPG" 

"whereas, 2h PPG was reduced": "whereas 2h PPG was reduced" 

"that included treatment-naive patients, showed": "that included treatment-naive patients showed" 

"a real-world setting": you might want to define this term the first time it appears in the text. Also please write consistently with a hyphen throughout the manuscript

"were significantly higher, when added": "were significantly higher when added" 

"After 24 weeks of intervention HbA1c...": "After 24 weeks of intervention, HbA1c..." 

"Empa 10mg and 25mg" : "Empa 10 mg and 25 mg" 

"On the top of that,": "On top of that," 

"Ertu is the most recent approved": "Ertu is the most recently approved" 

"After 26 weeks of intervention HbA1c..." : "After 26 weeks of intervention, HbA1c..." 

RAASi : please define acronym

"The CANVAS Program trial consisted from a combination": "The CANVAS Program trial consisted in a combination" 

"The estimated study completion date is on September 2019": "The estimated study completion date is September 2019" 

"The CV effects of SGLT2i apply also for individuals with T2DM and CKD [72]": "The CV effects of SGLT2i apply also to individuals with T2DM and CKD [72]" 

"while for major CV events the change was significantly only": "while for major CV events the change was significant only" 

"Nevertheless, it is unlikely that the latter fully explain": "Nevertheless, it is unlikely that the latter can fully explain" 

"Dyslipidemia is a common comorbidity of type 2 DM": "Dyslipidemia is a common comorbidity of T2DM" 

"which in another marker of endothelial function [99]": "which is another marker of endothelial function [99]" 

"In diabetic heart this metabolic": "In diabetic hearts, this metabolic" 

"overproduction of FFAs intermediates": "overproduction of FFA intermediates" 

"lead to a starvation situation": "lead to an starvation situation"

"and ketone bodies levels": "and ketone body levels" 

NHE isoform 1 : define NHE acronym

"A recent study by Timothy Garvey et al.": "A recent study by Garvey et al." 

"As discussed previously, SGLT2i lead to increased circulating ketone bodies": "As discussed previously, SGLT2i leads to increased circulating ketone bodies" 

type 1 DM : change to T1DM throughout the manuscript, define on first use

"Real world data": Change to "Real-world data" throughout the manuscript 

"In drug specific meta-analysis": "In drug specific meta-analyses"

"while the rest skeleton seems to remain uninfected": "while the rest of the skeleton seems to remain unaffected"

"a systematic review and meta-analysis, observed a non-significant difference in the risk of overall cancer": "a systematic review and meta-analysis observed a non-significant difference in the risk of overall cancer"

"urinary truck infections": "urinary tract infections" 

"An increased risk for hypoglycemia was however observed, when Empa, Cana or Dapa were added on treatment with insulin or SU": "An increased risk for hypoglycemia was however observed when Empa, Cana or Dapa were added on treatment with insulin or SU" 

"with an SGLT2i": "with a SGLT2i" 

"users vs. non-users patients": "user vs. non-user patients" 

"Nevertheless, the predominant pathophysiological mechanisms that may explain the CV 694 benefits of SGLT2i include plasma volume and diuresis, cardiac fibrosis, myocardial metabolism as 695 well as adipokine kinetics": "The predominant pathophysiological mechanisms that may explain the CV 694 benefits of SGLT2i include plasma volume and diuresis, cardiac fibrosis, myocardial metabolism as 695 well as adipokine kinetics" 

Reviewer 2 Report

This article is a very good and thorough review, considering the health and social issues involved with diabetes type II and the steady increase in its incidence worldwide, which is foreseen by all kinds of contemporary prognostic instruments. Further, although SGLT2 inhibitors have long since been around on the empirical basis to help control diabetes, it is just now that the specific molecular mechanisms of action are getting unraveled. My clinical experience is that this class of drugs is seldom used, much less than it should have been, due to lack of savvy in the mechanisms of action and potential benefits. Also, physicians shun using these drugs for fear of side effects. In this sense the review fills in the knowledge gap and is a desirable and timely piece of information.

Having said that I would like to remark that a perfect review also should point to the possible new ideas, new avenues of research and developments in the field discussed, let alone be as clinically useful as possible. To this end I miss some aspects and would like to suggest plausible improvements. The authors mention four drugs of this class. That is surprisingly a clinical problem limiting the use, doctors usually are unsure which one to use, so they skip all of them, moving to a more traditional class of drugs. Another thing is that each of the four comes in 2-3 tablet doses. Increasing the dose negligibly only increases the antidiabetic benefit. So why to use greater doses unless they would appreciably stronger counteract and protect from cardiovascular morbidity and mortality, at the same time not increasing the severity of adverse effects. Your article does not give a straight forward answer to that and I think, you could do better on that, at least in the conclusions.

Benefits in cardiovascular morbidity, mortality, hospitalizations, etc., but also in other metabolic, typical for diabetes, disorders, using these drugs, are so stunning that one wonders if and when SGLT2 inhibitors would substitute for metformin as the first line single treatment. Perhaps not, but why not, what would be the counter indication for such substitution.

On the more futuristic note, although at first glance beyond the main scope of your review, but in fact related to it, is the plausible use of SGLT2 inhibitors in type 3 diabetes, i.e., in brain tissue insulin resistance or dysfunctional GLUT3 receptors, in a diabetes of the aged not involving overweight and other metabolic alterations. Do the ingested SGLT2 inhibitors cross the blood-brain barrier, do they have any effect on glucose transport to the brain. It would be a good, highly relevant complement to your review, particularly in the face of rapidly aging societies and spiking glucose intolerance in senescence.

Lines 81 & 350 grammatical typos !
